# CrossMath: Towards Cross-lingual Math Information Retrieval

## ABSTRACT

Current math search engines and test collections are primarily developed for the English language, limiting their accessibility and inclusivity. This paper introduces cross-lingual math information retrieval (CLMIR) to overcome this limitation, focusing on retrieving mathematical information across languages. The paper presents CrossMath, a novel CLMIR test collection comprising manually translated topics in four languages (Croatian, Czech, Persian, and Spanish). Additionally, a CLMIR system leveraging state-of-the-art translation models (mBART and NLLB) alongside a formula masking approach to handle mathematical notation is introduced. Evaluation results on the ARQMath test collections show the effectiveness of the proposed CLMIR system, indicating competitive effectiveness against using English topics for all four languages.

## CCS CONCEPTS

• **Information systems → Mathematics retrieval**.

## KEYWORDS

Math IR, Cross-lingual Information Retrieval, Technical Documents

**ACM Reference Format:**

Anonymous Author(s). 2023. CrossMath: Towards Cross-lingual Math Information Retrieval. In *Proceedings of Make sure to enter the correct conference title from your rights confirmation emai (ICTIR).* ACM, New York, NY, USA, 5 pages. https://doi.org/XXXXXXX.XXXXXXX

## 1 INTRODUCTION

Math information retrieval (MathIR) is the specialized field concerned with finding relevant documents for queries expressed in mathematical language. Unlike traditional information retrieval, MathIR deals with a multimodal challenge: both user queries and the document collections they search through involve not only natural language text but also symbolic expressions in the form of mathematical formulas. This inherent complexity is further compounded by the technical nature of mathematics itself. Research suggests that users exhibit distinct search behaviors in the context of MathIR compared to general web searches. Studies have shown that users invest greater effort in formulating precise mathematical queries [8].

Despite these inherent challenges, the field of MathIR has witnessed continuous research efforts directed towards developing essential resources like test collections and effective search systems. A recent noteworthy initiative is the creation of the ARQMath

*ICTIR, July 13, 2024, Washington D.C., USA*

test collections [7, 9, 14]. Built upon the vast knowledge base of Math Stack Exchange, these collections serve the critical purpose of facilitating the evaluation of MathIR search systems. The core task within ARQMath focuses on retrieving relevant answers to user-posed mathematical questions.

Different math search engines have also been introduced in recent years, supporting text and formula input queries, such as Appraoch0 [15], MathDeck [4], and MathMex [5]. These search engines leverage vast repositories of mathematical knowledge, including resources like Wikipedia, Math Stack Exchange, and arXiv. However, a current limitation is their dependence on English-language queries. This restricts their accessibility to users who may be more comfortable formulating their problems in other languages.

In this paper, we introduce the problem of cross-lingual math information retrieval (CLMIR) as a specialized branch of cross-lingual information retrieval (CLIR) focused on retrieving mathematical information. We formally define the CLMIR problem in its simplest form: given a mathematical query formulated in any language, the task is to identify and return documents that address the query effectively. CLIR has been investigated in other domains such as legal [1], biomedicine [12], and e-commerce [10]. However, to the best of our knowledge, this has never been studied before for the math domain, and in this paper, we explore this problem.

We introduce CrossMath, a novel cross-lingual math information retrieval (CLMIR) test collection that expands upon the existing ARQMath test collection. CrossMath achieves this by manually translating ARQMath's topics into four additional languages: Croatian, Czech, Persian, and Spanish. This enriched dataset allows for a more comprehensive evaluation of CLMIR systems across a wider range of languages.

Then, we present a CLMIR system that leverages two state-of-the-art translation models, mBART [6] and NLLB (No Language Left Behind) [3]. These models face the challenge of translating mathematical queries while preserving their symbolic structure. To achieve this, we implement a masking approach that identifies and protects mathematical formulae during the translation process. Following translation, we utilize a Sentence-BERT model [11] for document retrieval, enabling the system to efficiently search for relevant documents based on the semantic similarity between the translated query and the content within the document collection.

We evaluate the effectiveness of our proposed CLMIR system on the ARQMath test collections using two standard metrics: {Precision′, nDCG′}@10. The evaluation results demonstrate that, for all four languages (Czech, Croatian, Persian, and Spanish), our proposed pipeline achieves competitive effectiveness close to the baseline model that utilizes original English topics. In summary, this work makes the following key contributions:

- CrossMath test collection: a novel cross-lingual math information retrieval (CLMIR) test collection that expands upon the existing ARQMath collection.
- CrossMath CLMIR system: capable of translating mathematical documents and retrieving relevant results.

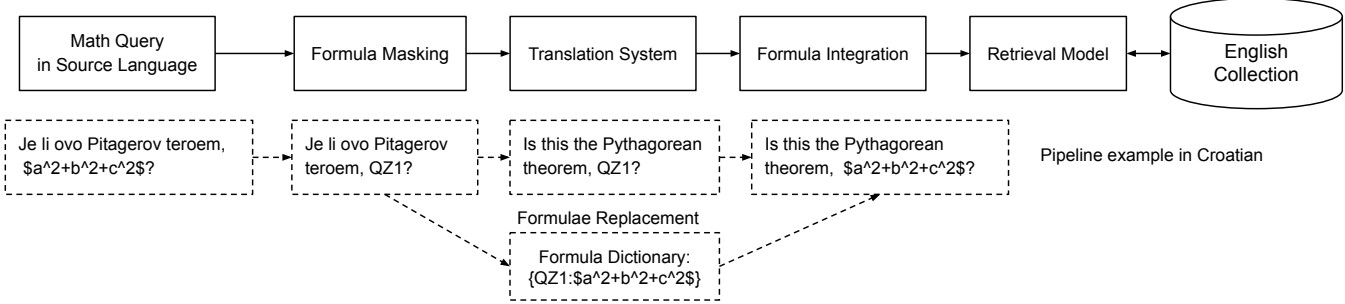

**Figure 1: CrossMath Search Pipeline. The dotted diagram shows an example in the Croatian language.**

## 2 CROSSMATH TEST COLLECTION

This section describes our approach for generating the dataset. We leverage the ARQMath test collections [7, 9, 14] as the foundation for our work. These collections utilize data from Math Stack Exchange. The primary task within ARQMath is math answer retrieval: given a mathematical question, the system aims to retrieve relevant answers. The searchable collection consists of answers provided to questions from 2010 to 2018. Topics (questions) for retrieval are drawn from those posted in 2019 (ARQMath-1), 2020 (ARQMath-2), and 2021 (ARQMath-3).

The three ARQMath test collections we utilize provide a total of 244 topics for the answer retrieval task. Each topic comprises a question title and body. To expand the collection for cross-lingual retrieval, we manually translated the topics into four additional languages: Croatian, Czech, Spanish, and Persian.[1] All assessors were native speakers with college-level mathematics backgrounds, ensuring accurate and domain-specific translations.

Both the title and body of each topic were translated into the four target languages. These translated topics serve as the input queries for CLMIR systems and are publicly available after the review process.[2] Figure 2 shows an example topic (A.4 from ARQMath-1) in the source language (English) alongside its translation into Persian as the target language.

## 3 CROSSMATH RETRIEVAL MODELS

This section details our CLMIR pipeline, visualized in Figure 1 using a Croatian language example. Given a math query in its source language, our model begins by extracting formulae. Each formula is then replaced with a unique identifier (special token) and stored in a formula dictionary for later reinsertion. Following this masking process, the text undergoes translation. Once translated, the system replaces the formula placeholders within the translated text with their corresponding original formulae retrieved from the dictionary. Finally, the processed query is fed into a retrieval model to identify relevant answers. The formula masker recognizes the following delimiters as formulae:

- \(...\)
- $...$ and $$...$$
- \[...\]
- \begin{...}...\end{...}

---

[1]Our choice of languages was bounded by the availability of assessors.
[2]To appear after review process

**English Topic**
<Topics>
...
<Topic number="A.4">
<Title>How to compute this combinatoric sum?</Title>
<Question> I have the sum $$\sum_{k=0}^{n}\binom{n}{k} k$$ I know
...
</Question>
</Topic>
...
</Topics>
**Persian Topic**
<Topics>
...
<Topic number="A.4">
<Title>چگونه این مجموعه ترکیبی را محاسبه کنیم؟</Title>
<Question>
یک سری $$\sum_{k=0}^{n}\binom{n}{k} k$$ دارم من می‌دانم...
</Question>
</Topic>
...
</Topics>

**Figure 2: An example ARQMath topic, A.4 in English and Persian.**

After the formulae are identified, the formulae masking is applied. We replace each formula, with a special token($QZ$) + an integer identifier (e.g., QZ1, QZ2, QZ3, ...). The special token for our experiment was chosen as $QZ$ for two reasons; A 2-character sequence is efficient to process, and the unlikelihood of confusion with existing language units. The original formula and the masked token are saved in a formula dictionary, which is used for reinsertion into the translated text later in the pipeline.

Masking the formulae serves to protect the semantics of a query. This is because *LaTeX* formulae may confuse and derail translation models. Additionally, if a small special token is chosen, masking will reduce the amount of characters. This reduced size will then increase the model's speed and efficiency. To demonstrate why we are masking, consider the following English body that was translated to Czech

```
"Let $S$ be a nonempty set of natural numbers. Is the
following formula $$ \exists p\ \bigl(\text{$p$ is prime}
\rightarrow \forall x \text{ ($x$ is prime)}\bigr)$$ ..."
```

**Table 1: Retrieval Results for English Language, on ARQMath-1 to -3 using P′@10 and nDCG′@10.**

| Ret. Model | NDCG′@10 | | | P′@10 | | |
|---|---|---|---|---|---|---|
| | ARQMath-1 | ARQMath-2 | ARQMath-3 | ARQMath-1 | ARQMath-2 | ARQMath-3 |
| Bi-encoder | 0.259 | 0.208 | 0.245 | 0.283 | 0.211 | 0.304 |
| Cross-encoder | 0.040 | 0.058 | 0.054 | 0.051 | 0.071 | 0.082 |

**Table 2: Retrieval Results for Croatian Language, on ARQMath-1 to -3 using P′@10 and nDCG′@10.**

| Translator | Ret. Model | NDCG′@10 | | | P′@10 | | |
|---|---|---|---|---|---|---|---|
| | | ARQMath-1 | ARQMath-2 | ARQMath-3 | ARQMath-1 | ARQMath-2 | ARQMath-3 |
| mBART | Bi-encoder | 0.227 | 0.161 | **0.221** | 0.256 | 0.175 | **0.272** |
| | Cross-encoder | 0.045 | 0.047 | 0.059 | 0.058 | 0.058 | 0.083 |
| NLLB | Bi-encoder | **0.232** | **0.187** | 0.205 | **0.258** | **0.191** | 0.247 |
| | Cross-encoder | 0.044 | 0.055 | 0.063 | 0.053 | 0.063 | 0.094 |

**Table 3: Retrieval Results for Czech Language, on ARQMath-1 to -3 using P′@10 and nDCG′@10.**

| Translator | Ret. Model | NDCG′@10 | | | P′@10 | | |
|---|---|---|---|---|---|---|---|
| | | ARQMath-1 | ARQMath-2 | ARQMath-3 | ARQMath-1 | ARQMath-2 | ARQMath-3 |
| mBART | Bi-encoder | **0.250** | 0.194 | 0.228 | **0.270** | 0.199 | 0.283 |
| | Cross-encoder | 0.035 | 0.059 | 0.052 | 0.055 | 0.072 | 0.081 |
| NLLB | Bi-encoder | 0.240 | **0.200** | **0.233** | 0.262 | **0.202** | **0.294** |
| | Cross-encoder | 0.030 | 0.060 | 0.057 | 0.045 | 0.069 | 0.085 |

was transformed into `"Zvláštní logický vzorec zahrnující prvorčísla 00:47:53:53 00:44:53:53:00:44:53:53:53:53: 53:54 00:44:53:53:53:53:53:53 ..."` when no masking is applied. The erroneous sequence disappears once the sentence is masked and fed through the translation pipeline, coming out as `"Zvláštní logický vzorec zahrnující prvorčísla Ať je $S$ neprázdný soubor přirozených čísel. Je následující vzorce $$ \exists p\ \bigl(\text{$p$ is prime} \rightarrow \forall x \text{($x$ is prime)}\bigr) $$ ..."`

The formulae-masked text is then passed to a translation model. In our work, we considered two models: mBART [6] and NLLB [3]. While mBART can be a good choice for high-resource languages, we consider NLLB as an alternative to study its effectiveness for low-resource languages like Persian. After translation, using the formula dictionary, the masked formulae are reinserted in the text, forming the input query for retrieval models.

Our CLMIR pipeline employs two retrieval models based on Sentence-BERT [11]. The first leverages a bi-encoder architecture, specifically the 'all-mpnet-base-v2' model, recognized for its quality among pre-trained bi-encoders. In this model, both the query and candidate documents are processed, and their cosine similarity based on vector representations determines the relevance score. The second retrieval model utilizes a cross-encoder architecture, the 'qnli-distilroberta-base' model. This model was trained on QNLI [13] for the task of determining whether a given question can be answered by a Wikipedia passage. The query and the candidate document are passed to the model, and their relevance score is predicted. Note that both models are used without any fine-tuning. Fine-tuning these models with ARQMath data has the potential to further enhance retrieval results, which we explore as future work.

## 4 EXPERIMENTAL SETTING

We leverage the ARQMath-1, -2, and -3 test collections for evaluation. These collections provide math questions in English, each with a title and body. We utilize the translated versions of these topics, specifically the concatenated title, and body, as input queries for our CLMIR pipeline.

For each topic in ARQMath, answers are assessed with four relevance degrees: high, medium, low, and non-relevant. We re-ranked the assessed answers for each topic (on average 468 answers per topic). We used P′@10[3], and nDCG′@10, using Python ranx library [2]. Following ARQMath evaluation protocols, when calculating P′@10, we considered only medium and high as relevant. For each language, there are four retrieval results; two translation models, and two retrieval models for each translation. We applied a two-sided paired student's t-test with $p = 0.05$ to compare retrieval results in our experiments. As the baseline model, we compare our results using both retrieval models in English.

## 5 EXPERIMENTAL RESULTS

This section provides the retrieval results and analysis. Using two translation models, and two retrieval models, the results for each language are shown in the following tables: Table 2 for Croatian, Table 3 for Czech, Table 4 for Persian, and Table 5 for Spanish.

Our observations align with the findings for the English baseline. Across all languages, the bi-encoder retrieval model significantly outperformed the cross-encoder model. Interestingly, when employing the bi-encoder model, no statistically significant difference was observed in the effectiveness between queries translated with mBART or NLLB. However, for Persian and Spanish, the NLLB

---

[3]Prime denoting that we are re-ranking only the assessed documents

**Table 4: Retrieval Results for Persian Language, on ARQMath-1 to -3 using P′@10 and nDCG′@10.**

| Translator | Ret. Model | NDCG′@10 | | | P′@10 | | |
|---|---|---|---|---|---|---|---|
| | | ARQMath-1 | ARQMath-2 | ARQMath-3 | ARQMath-1 | ARQMath-2 | ARQMath-3 |
| mBART | Bi-encoder | 0.228 | 0.183 | 0.199 | 0.248 | 0.185 | 0.245 |
| | Cross-encoder | 0.052 | 0.054 | 0.065 | 0.060 | 0.058 | 0.097 |
| NLLB | Bi-encoder | **0.253** | **0.191** | **0.213** | **0.270** | **0.185** | **0.251** |
| | Cross-encoder | 0.065 | 0.051 | 0.054 | 0.068 | 0.056 | 0.082 |

**Table 5: Retrieval Results for Spanish Language, on ARQMath-1 to -3 using P′@10 and nDCG′@10.**

| Translator | Ret. Model | NDCG′@10 | | | P′@10 | | |
|---|---|---|---|---|---|---|---|
| | | ARQMath-1 | ARQMath-2 | ARQMath-3 | ARQMath-1 | ARQMath-2 | ARQMath-3 |
| mBART | Bi-encoder | 0.232 | 0.159 | 0.189 | 0.244 | 0.166 | 0.249 |
| | Cross-encoder | 0.044 | 0.061 | 0.068 | 0.053 | 0.065 | 0.096 |
| NLLB | Bi-encoder | **0.256** | **0.195** | **0.243** | **0.278** | **0.196** | **0.299** |
| | Cross-encoder | 0.045 | 0.062 | 0.058 | 0.058 | 0.078 | 0.083 |

translation model yielded better results across all ARQMath test collections. In contrast, for Czech and Croatian, NLLB achieved higher effectiveness in two out of the three test collections.

Our observations suggest that NLLB might be better suited for translating mathematical concepts. Consider a topic that begins with the sentence: "Show that $\sqrt{n}$ is irrational if $n$ is not a perfect square, using the method of infinite descent.", The mBART translation into Persian renders this as: "Show that $\sqrt{n}$ is square if $n$ is not a square, using the infinite depreciation method." In contrast, NLLB translates it as: "Indicate that $\sqrt{n}$ is indivisible if $n$ is not a perfect square, using the method of infinite descent." For this specific topic, the P′@10 value increases from 0.1 to 0.5 when using NLLB instead of mBART, highlighting the potential benefit of NLLB for certain mathematical domains and languages.

Our experiments reveal that while no language surpassed the effectiveness achieved with the original English topics, there was no statistically significant difference between the performance of English searches and other languages when using the bi-encoder retrieval model with NLLB translations. This suggests that our CLMIR pipeline can achieve comparable performance for certain languages, even with the additional challenge of translation.

Analyzing the effectiveness by the difficulty of topics (ARQMath labels), Figure 3 shows the P′@10 for each language across each difficulty level. Notably, for English, the P′ @10 values for medium and easy topics are very similar. In contrast, languages like Czech and Persian exhibit higher effectiveness for medium-difficulty topics. Conversely, Croatian demonstrates a decrease in effectiveness for medium topics compared to easy ones.

Upon closer examination, we observed that medium-difficulty topics sometimes contained terms that the translation models struggled to translate accurately. These terms often corresponded to crucial keywords within the questions. For instance, the topic titled 'Inequality between norm 1, norm 2 and norm ∞ of Matrices' was translated into Croatian as 'Nejednakost između norme 1, norme 2 i norme ∞ matričnih.' When translated to English with NLLB, the two important words (Nejednakost: Inequality) and (matričnih: Matrices) remained untranslated. Interestingly, for Persian, omitting certain keywords like 'covariance' and 'independent' in a topic related to random variables actually improved effectiveness. The model assigned higher ranks to more general and relevant topics

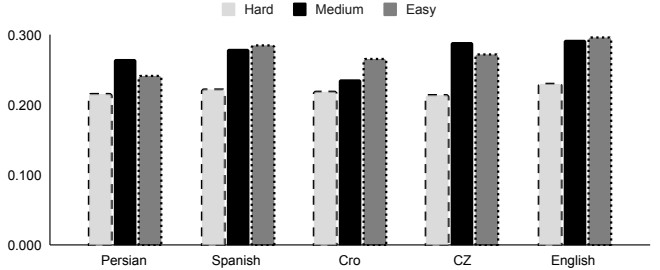

**Figure 3: P′@10 per topic complexity for each language.**

on random variables, suggesting the potential benefits of keyword management strategies for specific languages. In conclusion, our analysis reveals that the CLMIR system, particularly when employing the bi-encoder search model, demonstrates consistent behavior across all ARQMath test collections and various topic categories, mirroring the effectiveness observed with English topics. However, the impact of topic complexity highlights the potential need for further exploration of advanced translation techniques to ensure accurate capture of crucial mathematical concepts.

## 6 CONCLUSION

This paper introduces CrossMath, the first cross-lingual math information retrieval (CLMIR) test collection and retrieval model. Leveraging the existing ARQMath collection, CrossMath expands the scope of MathIR by providing topic translations in four languages: Croatian, Czech, Persian, and Spanish. We propose a CLMIR pipeline that addresses the challenges of formula preservation during translation. The pipeline employs formula masking, followed by translation and formula reinsertion. Our experiments demonstrate that the retrieval effectiveness achieved with the bi-encoder search model in our CLMIR system is comparable to using the original English topics. For future work, we plan to investigate the effectiveness of fine-tuning translation models specifically for mathematical language. Additionally, we aim to enrich CrossMath by incorporating a wider range of languages, encompassing both high-resource languages like Chinese and low-resource languages like Somali.

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
