# OpenReview forum: "CrossMath: Towards Cross-lingual Math Information Retrieval"
_ACM.org/SIGIR/ICTIR/2024/Conference — ICTIR 2024_

### Official Review · Reviewer_z8Z2 · 2024-05-03

**Rating:** 1
**Confidence:** 4

**Objective Part Of Review:**

The problem, method, and results of cross-lingual math information retrieval are clearly stated. Several writing problems:

- Figure 2 is mentioned earlier in the text than Figure 1, which may not be in the right order.
- I would suggest using the full language name in Figure 3. No abbreviation is needed here since the space is enough.

---

For the method, did you try to use a multi-lingual retriever like [mDPR](https://github.com/mia-workshop/MIA-Shared-Task-2022/tree/main/baseline/mDPR) to directly perform end-to-end retrieval rather than relying on the translator?

---

For the result, I would like to see more intrinsic metrics to measure the translation accuracy, whatever the human evaluation or automatic BLEU and ROUGE scores with the ground truth.

**Subjective Part Of Review:**

The overall writing is clear and easy to understand. Cross-lingual information retrieval in the math domain has not been well explored. This paper first introduces the ARQMath dataset translated into 4 different languages and then proposes an end-to-end translation + retrieval pipeline. The cross-lingual results can match the original English ones, showing their method's effectiveness. The dataset and method in this paper may have a positive impact on math search users across the world. Fine-tuning translation models/retrievers specifically for mathematical language can be an interesting topic and follow-up work.

---

### Official Review · Reviewer_CoAH · 2024-05-09

**Rating:** -1
**Confidence:** 4

**Objective Part Of Review:**

Generally, the paper is well written and easy to understand (e.g., problem settings, methodology and results are clearly stated). The introduction provides clear outlines of the motivation and approaches to readers

**Subjective Part Of Review:**

In my opinion, the biggest contribution of this paper is to provide the manually translated math queries from ARQMath dataset to the community. The proposed approach (baseline) is intuitive but the biggest concern in this paper for me is that as the first paper to address cross-lingual math search, there should be more baselines to explore. (1) Using cross-encoder fine-tuned on relevance judgement rather than NLI (e.g., cross-encoder/ms-marco-MiniLM-L-6-v2). The reported numbers for cross-encoder are much lower than bi-encoder, which is surprised to me. However, the used bi-encoder and cross-ender models are fine-tuned in totally different data, which might be the reason. (2) Using existing multilingual dense retrievers (e.g., facebook/mcontriever-msmarco). Although it makes sense that translated queries into the target language may perform well, to make the experiment comprehensive, directly using multilingual dense retrieval is a reasonable baseline. (3) The proposed translation with masked equations seems reasonable but it also prompts readers to think how much the performance would degrade if we directly translate the queries without masked equations.

---

### Official Review · Reviewer_yiW6 · 2024-05-12

**Rating:** -1
**Confidence:** 3

**Objective Part Of Review:**

The submission effectively states the problem of cross-lingual math information retrieval (CLMIR), highlighting the constraints posed by currently available English-centric math search engines and demonstrating the necessity for robust multilingual support. The introduction of a new dataset tailored for this purpose is a valuable contribution, offering a solid foundation for future research.

The paper is well-written with clear explanation of the proposed model. However, there are some details and experiments missing in the experimental setup.

**Subjective Part Of Review:**

The paper was easy to read and understand. The problem can be relevant to ICTIR, in the topic of search systems.

The method is standard but the paper needs more baseline comparison and more result-based analysis. My major concern is regarding the system evaluation. The evaluation could be strengthened by including comparisons with simpler, more accessible baseline systems such as a comparison with direct translations performed by the Google Translate API (without masking) could provide a clearer perspective on the performance of the proposed models, especially given that they are not fine-tuned.

---

### Official Review · Reviewer_wZ3x · 2024-05-16

**Rating:** -1
**Confidence:** 3

**Objective Part Of Review:**

This paper focuses on a particular task: Cross-lingual Math Information Retrieval. Following the translate-then-retrieve pipeline, this work proposes a masking-revealing approach (named CrossMath) to improve the quality of translation, which later leads to better retrieval performance. The idea is easy to follow, and the experiential results (Table 2 and 3) demonstrate the effectiveness of CrossMath.

**Subjective Part Of Review:**

**Strengths**

This work generally draws attention to structured text (e.g., scientific formulas, tables, special characters) in neural machine translation models. The paper shows that with masking-revealing approach, the translation quality of queries with math formula can be greatly improved.

**Weaknesses**

However, skipping (masking-revealing) them is only considered in the **translation step**. The paper does not address how mathematical formulas should be effectively represented and utilized within **retrieval models** for query-document matching. In Figure 1, it is shown that after the integration of a formula, the retrieval model processes the query by tokenizing it in a manner that does not differentiate mathematical formulas from natural language text. For translation tasks, since math language is universal, skipping them is straightforward. But in retrieval tasks, math formulas in queries are crucial info and should be used to match formulas in documents. Contextualized representation works for natural language can backfire on math formulas. For example, suppose the query is $a \ge b$, what ranking score should $a \le b$ be? How about $\alpha \ge \beta$ where $a, b$ and $\alpha, \beta$ carry the same meaning, respectively? I think this paper didn't address the key challenges created by math formulas in neural retrieval models.

Moreover, there are other studies focusing on improving the translation quality of mathematical text which could be use as the baseline in this work.
* Ohri A, Schmah T. Machine translation of mathematical text. IEEE Access. 2021.
* Petersen F, Schubotz M, Greiner-Petter A, Gipp B. Neural machine translation for mathematical formulae. ACL 2023.

---

### Meta-Review · Area_Chair_J7yu · 2024-05-29

**Recommendation:** Accept (Oral)
**Confidence:** 4

**Metareview:**

This is a meta-review. The paper addresses cross-lingual math retrieval. The authors propose a new test collection for the task and a method based on formula masking. The reviews are a bit mixed, but leaning towards rejection. In the discussion phase, I tried to collect the main reasons to reject the paper. Based on that, two reviewers updated their review.

The reviewers provided some feedback and suggestions that can be used to improve the final version of the paper. Reviewer CoAH lists 3 baselines that can be considered (I checked this in the discussion phase and the reviewer intended the 3 suggestions to be 3 baselines). I interpreted the review of yiW6 as a weak accept rather than a weak reject because they do not provide reasons to reject the paper. In the discussion phase, the reviewer mentioned that an ablation study is needed to see the effectiveness of the model's modules. Reviewer wZ3x provides 2 additional references that can be used as baselines.

One important aspect that I took into account for my recommendation, is the fact that this is a short paper. Although ICTIR does not distinguish short papers and has only one maximum length (9 pages), I feel that we could evaluate it as a short paper and asking for more analysis is not appropriate. In my view, the contributions of the paper are good, esp. the test collection. I also appreciate the focus on other languages than English; I think this is a topic that we should encourage in the IR community. This paper could make a nice discussion in the poster session at ICTIR.